# Reducing Power Line Interference from sEMG Signals Based on Synchrosqueezed Wavelet Transform

**DOI:** 10.3390/s23115182

**Published:** 2023-05-29

**Authors:** Jingcheng Chen, Yining Sun, Shaoming Sun, Zhiming Yao

**Affiliations:** 1Institute of Intelligent Machines, Hefei Institutes of Physical Science, Chinese Academy of Sciences, Hefei 230031, China; cjc324@mail.ustc.edu.cn (J.C.);; 2University of Science and Technology of China, Hefei 230026, China; 3Chinese Academy of Sciences (Hefei) Institute of Technology Innovation, Hefei 230088, China; 4School of Mathematics and Computer, Tongling University, Tongling 244061, China

**Keywords:** power line interference, synchrosqueezed wavelet transform, surface electromyography, adaptive ridge extraction

## Abstract

Power line interference (PLI) is a major source of noise in sEMG signals. As the bandwidth of PLI overlaps with the sEMG signals, it can easily affect the interpretation of the signal. The processing methods used in the literature are mostly notch filtering and spectral interpolation. However, it is difficult for the former to reconcile the contradiction between completely filtering and avoiding signal distortion, while the latter performs poorly in the case of a time-varying PLI. To solve these, a novel synchrosqueezed-wavelet-transform (SWT)-based PLI filter is proposed. The local SWT was developed to reduce the computation cost while maintaining the frequency resolution. A ridge location method based on an adaptive threshold is presented. In addition, two ridge extraction methods (REMs) are proposed to fit different application requirements. Parameters were optimized before further study. Notch filtering, spectral interpolation, and the proposed filter were evaluated on the simulated signals and real signals. The output signal-to-noise ratio (SNR) ranges of the proposed filter with two different REMs are 18.53–24.57 and 18.57–26.92. Both the quantitative index and the time–frequency spectrum diagram show that the performance of the proposed filter is significantly better than that of the other filters.

## 1. Introduction

Surface electromyography (SEMG), which reflects muscle activation by detecting the cumulative potential on the skin’s surface caused by activation of local motion units, is widely used in fields such as human–machine interaction [1,2,3], human activity recognition [4,5,6], diagnosis of neuromuscular diseases [7,8,9], supervision of health status [10,11,12], and research on motor ability development [13,14,15]. One obstacle is that the raw sEMG signal is susceptible to noise and may have a low signal-to-noise ratio (SNR) [16,17,18,19]. Specifically, due to the discrepancy in the volume conductor of body tissue and the degree of activation, a complex coherent state is shown on the skin’s surface after the aliasing of electrical activities caused by the activation of different motor units. Due to this defect of the sEMG signal, the sensor data have a low amplitude, wide bandwidth (20–450 Hz), and non-stationary time sequence.

The most common sEMG noises include motion artifacts, crosstalk of other bioelectrical signals, white noise, and power line interference (PLI). Because the frequency range of motion artifacts and bioelectric crosstalk is lower than that of the sEMG signal, high-pass filtering can be used for processing. A high SNR and uniform low-intensity frequency distribution are usually present as white noise, which makes it reasonable to use the threshold denoising methods based on wavelet transform [20,21,22], empirical mode decomposition [23,24,25], and other technologies for processing. However, the frequency spectrum of PLI overlaps with that of the sEMG signal, and the energy of PLI is difficult to ignore or may even drown out the desired signal due to the change in the application environment and the user operation level. Thus, removing PLI from sEMG while reducing signal distortion is a challenge.

At present, a variety of PLI filtering techniques for sEMG have been proposed in the literature. The most common one is notch filtering [26,27,28], but this ignores the problem of signal distortion near the frequency of PLI. Specifically, it is difficult for a narrow-pass band to cope with low SNR or time-varying noise, while a wide-pass band is accompanied by large signal distortion. Alternative methods are adaptive filtering [26,29], spectral interpolation [27,30], and the matching pursuit algorithm [27,31]. However, the former requires an external reference signal, while the PLI signal is modeled as a time-invariant cosine function when using the latter two methods. However, the real PLI is a time-varying signal due to the fluctuation in the power supply, change in electrode contact impedance caused by perspiration or loosening, and electromagnetic interference caused by the change in the space environment of the human body. A typical signal of sEMG mixed with strong PLI is shown in Figure 1a, while Figure 1b shows the time-spectrum diagram using a short-time Fourier transform (STFT). The time–frequency spectrum of sEMG is similar to that of the non-uniform noise signal covering a wide bandwidth, while the PLI and its higher harmonics are the time-varying ridge near corresponding frequencies. In terms of the time–frequency spectrum diagram, there are obvious morphological differences between sEMG and PLI. Therefore, it is hypothesized that the component extraction technique based on time–frequency analysis can be applied to the filtering of PLI in the sEMG signal.

Time–frequency analysis, which models non-stationary signals as two-dimensional functions of time and frequency, is an important method for dealing with time-varying signals. Presently, component retrieval technology based on time–frequency analysis is widely used in various fields, such as fault analysis [32,33], seismic signal analysis [34,35], oscillation analysis [36], and signal separation [37]. In addition, researchers have also applied this to bio-signal processing. For instance, Sharma [38] introduced a wavelet-transform (WT)-based multicomponent retrieval method [39] to remove the PLI from ECG signals. In terms of sEMG, Zivanovic [40] developed a low-rank matrix factorization approach based on STFT to remove the harmonic and baseline noise from sEMG signals.

The commonly used time–frequency analysis methods include STFT, s-transform, Wigner–Ville distribution, WT, and empirical-mode-decomposition (EMD)-based methods. Among these, a fixed window function is used in STFT, which is only suitable for analyzing signals with roughly similar scales, while WT can adaptively adjust the time–frequency window, so is appropriate for analyzing the local characteristics of signals. However, limited by the principle of indeterminacy, the compromise between the time resolution and the frequency resolution is inevitable, which leads to the reduction in the time–frequency resolution and the difficulty in precisely separating signal components. In comparison, the Wigner–Ville distribution can improve the aggregation of time–frequency energy, but it is seriously affected by the cross-interference. The rearrangement technique not only enhances the time–frequency energy aggregation, but also restrains the cross-interference. Nevertheless, the signal cannot be reconstructed accurately through inverse transformation, which limits the practical application of this method. As EMD is self-adaptive, and pre-selecting the appropriate mother wavelet and optimized wavelet parameters are not needed, it is widely used to analyze non-stationary signals [41,42,43]. Moreover, advanced methods based on EMD, such as EEMD [44,45] and VMD [46], have been developed to deal with different problems. However, PLI is a signal that varies over a very narrow bandwidth around a predicted frequency, and it is not easy for EMD-based methods to provide a local decomposition that focuses on arbitrarily narrow bandwidth information, which limits the resolution of PLI filtering in the sEMG signal. To solve the above problems, Thakur [47] proposed the synchrosqueezed wavelet transform (SWT) based on continuous wavelet transform (CWT), and completed the mathematical proof of the compression, decomposition, and reconstruction process of SWT. SWT can redistribute the energy of each scale to its frequency center of gravity, so as to improve the time–frequency energy concentration, and preserve the reconstruction ability of the inverse transformation.

In this paper, a novel method for reducing PLI from the sEMG signal is proposed. To be specific, the local SWT is proposed to decompose the local time–frequency spectrum of the signal, and then an adaptive ridge location method (RLM) is used to extract the PLI component from the corresponding frequency band. Finally, the filtering process is completed by subtracting the ridge reconstruction data from the sEMG signal. The adaptive RLM algorithm based on SWT is presented in Section 2. First, the CWT and SWT algorithms are reviewed. Then, the ridge location method using an adaptive threshold is illustrated. In addition, two ridge extraction methods are proposed, namely, complete removal and compensation by the matrix completion technique. Section 3 describes the methods, and the quantitative evaluation parameters and time–frequency spectrum graph are used to evaluate the filtering effect of the proposed algorithm in simulated and real sEMG signals, respectively. Afterwards, the results are shown in Section 4, and the comparison between the proposed method and the conventional PLI filtering methods is given. Finally, the contribution and limitations of this paper are discussed and concluded in Section 5.

## 2. Description of the PLI Filters

The SWT-based PLI filter is described in this section. The flowchart of this method is shown in Figure 2. First, the local SWT is executed to fetch the neighborhood and PLI information with a preset resolution. Then, using the thrice standard error principle, the adaptive threshold is calculated from the neighborhood information. Based on this threshold, a RLM is carried out to adaptively locate precise PLI ridges from PLI information. Two REMs are developed to meet the requirements of two different applications, namely, direct elimination for faster and coarser requirements, and matrix completion for slower and more precise requirements. Finally, the PLI signal is estimated using the inverse transform of SWT (ISWT).

### 2.1. Continuous Wavelet Transform

For $x(t)\in L^{2}$ the CWT is defined as:(1)$$W_{x}\left(a,b\right)=\int_{-\infty }^{+\infty }x\left(t\right)\bar{\Psi_{a,b}\left(t\right)}dt$$
where *a* is the scaling factor and *b* is the shift factor, and $\bar{\left(\cdot \right)}$ is the complex conjugation operator. $\Psi_{a,b}\left(t\right)=\frac{1}{\sqrt{a}}\varphi \left(\frac{t-b}{a}\right)$, $\varphi \left(t\right)$ is the mother wavelet function, which satisfies $\int_{-\infty }^{+\infty }\frac{{\left|\hat{\varphi }\left(\xi \right)\right|}^{2}}{\left|\xi \right|}d\xi <+\infty$, where $\hat{\varphi }\left(\xi \right)$ is the Fourier transform of $\varphi \left(t\right)$.

According to the convolution theorem, the CWT in frequency domain can be defined as $\hat{W_{x}}\left(a,\xi \right)=\hat{x}\left(\xi \right)\cdot \sqrt{a}\bar{\hat{\varphi }\left(a\xi \right)}$. Thus, to reduce the computational complexity, Equation (1) can be converted to $W_{x}\left(a,\cdot \right)=F^{-1}\left(\hat{x}⨀\sqrt{a}\bar{\hat{\varphi }\left(a\cdot \right)}\right)$, with $F^{-1}$ as the inverse CWT operator.

The discretization of CWT in the scale domain is realized by a series of discrete scale factors, as $a_{k}=2^{\frac{k}{n_{v}}}△t,k=1,\cdots ,Ln_{v}$, where △t is the time span and $L={log}_{2}\frac{N^{\prime }}{2}$, $N^{\prime }$ is the smallest power of two not less than the length of the signal. $n_{v}$ is a user-defined parameter that affects the total number of scales [47]. Accordingly, ξ represents samples in the unit frequency interval, as $\xi_{i}=\frac{2\pi i}{N^{\prime }},i=0,\cdots ,N^{\prime }$ − 1.

### 2.2. Synchrosqueezed Wavelet Transform

Although the energy of the CWT is leaked in the scale direction, its phase is unchanged, so the original frequency information can be reflected by the oscillation in the time direction. Therefore, the instantaneous frequency can be calculated as:(2)$$\omega_{x}\left(a,b\right)=\frac{\partial_{b}W_{x}\left(a,b\right)}{{2\pi jW}_{x}\left(a,b\right)}$$
where *j* is the imaginary unit. Then, the wavelet coefficients in the scale-time domain are mapped to the frequency–time domain by synchrosqueezing:(3)$$T_{x}\left(\xi_{l},b\right)=\sum_{\left\{a_{k}:\left|\omega_{x}\left(a_{k},b\right)-\xi_{l}\right|\leq \frac{△\xi_{l}}{2}\right\}}W_{x}\left(a_{k},b\right){a_{k}}^{-\frac{3}{2}}△a_{k}$$
where $△\xi_{l}=\xi_{l}-\xi_{l-1}$, $△a_{k}$=$a_{k}-a_{k-1}$.

The ISWT is:(4)$$x\left(t\right)=Re\left[\frac{1}{C_{\psi }}\sum_{l}T_{x}\left(\xi_{l},b\right)\right]$$
where $C_{\psi }=\int_{0}^{+\infty }\frac{\bar{\hat{\varphi }\left(\xi \right)}}{\xi }d\xi$.

### 2.3. Local CWT/SWT Related to the PLI

In fact, it is redundant to calculate the wavelet coefficients at all scales, as most of them do not contain any information related to the PLI. Therefore, it is very beneficial to reduce the computational complexity by calculating only the local CWT (LCWT) and local SWT (LSWT) related to the PLI signals.

Firstly, the target frequency sets for filtering is defined as:(5)$$Ta=\left\{\tilde{\xi_{l}}|\xi_{c}-\xi_{w}\leq \tilde{\xi_{l}}\leq \xi_{c}+\xi_{w}\right\}$$
where $\xi_{c}$ is the estimated center frequency for filtering, and $\xi_{w}$ is half of the filtering bandwidth. The neighbourhood sets of Ta is defined as: (6)$$Ne=\left\{\tilde{\xi_{l}}|\xi_{c}-3\xi_{w}\leq \tilde{\xi_{l}}<\xi_{c}-\xi_{w}\;or\;\xi_{c}+\xi_{w}<\tilde{\xi_{l}}\leq \xi_{c}+3\xi_{w}\right\}$$

Then, the local sets of frequency Lo=Ta∪Ne.

Consider the set of scales that can fully explain the frequency information corresponding to set Lo. Due to the leakage of the wavelet coefficients in the scale domain, the frequency information mapped by $a_{k}$ has a bandwidth of $\left[\frac{\left(F_{c}-F_{w}\right)F_{s}}{a_{k}},\frac{\left(F_{c}+F_{w}\right)F_{s}}{a_{k}}\right]$, with $F_{s}$ as the sampling frequency, while $F_{c}$ and $F_{w}$ are the center frequency and half of the bandwidth of the mother wavelet function, respectively. The scale sets that adequately cover Lo are defined as:(7)$$C=\left\{a_{k}|m\leq k\leq M,\frac{\left(F_{c}+F_{w}\right)F_{s}}{a_{m}}\geq \xi_{c}-3\xi_{w},\;and\;\frac{\left(F_{c}-F_{w}\right)F_{s}}{a_{M}}\geq \xi_{c}+3\xi_{w}\right\}$$

Obviously, there is more than one set that satisfies Equation (7), and $C_{c}$ with the smallest cardinality (#C) is chosen to simplify the computation, namely, the compact scale set. Finally, LCWT and LSWT are calculated in $C_{c}$ and Lo, respectively.

### 2.4. Value of $n_{v}$

Usually, the value of $n_{v}$ is assigned as an empirical value, such as 32 or 64 [39,47]. However, it is not appropriate when dealing with multiple separate filtering target frequencies (e.g., power line frequency and its higher harmonics) to use a constant $n_{v}$. Considering that $\tilde{\xi_{l}}$ is uniformly distributed in set Lo according to the resolution $\xi_{r}$, it is expected that the cardinality of $C_{c}$ satisfies $\#C_{c}\approx \frac{\left(\xi_{c}+3\xi_{w}\right)-\left(\xi_{c}-3\xi_{w}\right)}{\xi_{r}}$. It is known that $\frac{\xi_{c}+3\xi_{w}}{\xi_{c}-3\xi_{w}}\approx \frac{\xi_{m}}{\xi_{M}}=2^{\frac{\#C_{c}}{n_{v}}}$; then:(8)$$n_{v}=minimum(n_{i}\in N^{+}|n_{i}\geq \frac{{6\xi }_{w}}{\xi_{r}\log_{2}\frac{\xi_{c}+3\xi_{w}}{\xi_{c}-3\xi_{w}}})$$

### 2.5. Adaptive Threshold

The spectrum of raw sEMG signals tends to be distributed continuously. Therefore, the spectrum within a certain bandwidth can be estimated from the information of its neighborhood bandwidth, which is also applicable to the time–frequency spectrum of SWT. Based on this, the adaptive threshold is defined by the thrice standard error principle as:(9)$$\sigma_{b}=\frac{\sum_{l\in N_{e}}T_{x}\left(\tilde{\xi_{l}},b\right)}{\#Ne}+3{\left(\frac{\sum_{l\in N_{e}}{\left(T_{x}\left(\tilde{\xi_{l}},b\right)-\frac{\sum_{l\in N_{e}}T_{x}\left(\tilde{\xi_{l}},b\right)}{\#Ne}\right)}^{2}}{\#Ne}\right)}^{0.5}$$

### 2.6. RLM Algorithm

The energy aggregation and sparsity of the time–frequency spectrum of the signal are enhanced after the SWT is carried out, which leads to the main information defined by Lo consisting of several time–frequency spectrum peaks. The peaks are defined as:(10)$$H_{k}\left(\gamma ,b\right)=\left\{\begin{array}{c} l^{k}=l_{B}^{k},\cdots ,l_{E}^{k},E>B:\forall l_{i}^{k}\in l^{k},\left|T_{x}\left(\tilde{\xi_{l_{i}^{k}}},b\right)\right|>\gamma ,\\ and\;\left|T_{x}\left(\tilde{\xi_{l_{B-1}^{k}}},b\right)\right|\leq \gamma ,\left|T_{x}\left(\tilde{\xi_{l_{E+1}^{k}}},b\right)\right|\leq \gamma \end{array}\right\}$$

The peaks’ values are defined as:(11)$$mH_{k}\left(\gamma ,b\right)=maximum\left(\left|T_{x}\left(\tilde{\xi_{l^{k}}},b\right)\right|\right),l^{k}\in H_{k}\left(\gamma ,b\right)$$

The method for ridge location is proposed as follows:

First, the center of gravity of the ridge $l_{b}^{max}$ in *b* is defined as the location that has the maximum value of $T_{x}\left(\tilde{\xi_{l}},b\right)$. Then, the auxiliary set for ridge location is defined as:(12)$$ARL\left(b\right)=\left\{l|minimum\left({ARL}^{*}\left(b\right)\right)\leq l\leq maximum\left({ARL}^{*}\left(b\right)\right)\right\}\;{ARL}^{*}\left(b\right)=\left\{\begin{array}{c} l|l\in H,\;l_{b}^{max}\in H,H=\left\{H_{B}⋃\cdots ⋃H_{E}\right\}\in C,E>B,\;s.t.\forall H_{i}\in H,mH_{i}>\sigma_{b},\\ and\;\left[mH_{B-1}\leq \sigma_{b}\;or\;H_{B-1}\notin C\right],\;\left[mH_{E+1}\leq \sigma_{b}\;or\;H_{E+1}\notin C\right] \end{array}\right\}$$

Finally, the ridge in *b* is located at:(13)$$RL\left(b\right)=\left\{\begin{array}{c} l=l_{B},\cdots ,l_{E},E>B,l\in ARL\left(b\right):\left|T_{x}\left(\tilde{\xi_{l_{B}}},b\right)\right|>\sigma_{b},\left|T_{x}\left(\tilde{\xi_{l_{E}}},b\right)\right|>\sigma_{b},\\ and\;s.t.\forall l^{*}\in \left(ARL\left(b\right)-RL\left(b\right)\right),\left|T_{x}\left(\tilde{\xi_{l^{*}}},b\right)\right|\leq \sigma_{b} \end{array}\right\}$$

In particular, $RL\left(b_{i}\right)$ is set to Null when $maximum\left(T_{x}\left(\cdot ,b_{i}\right)\right)\leq \sigma_{b_{i}}$.

### 2.7. Ridge Extraction Methods for PLI Filtering

The PLI components can be extracted by two possible methods. In the first method (REM1), the PLI signal is estimated by ISWT of the local SWT defined by the ridge location, as:(14)$$T_{x}^{REM1}\left(\tilde{\xi_{l}},b_{i}\right)=\left\{\begin{array}{l} T_{x}\left(\tilde{\xi_{l}},b_{i}\right),\;\;l\in RL\left(b_{i}\right)\\ 0,\;\;\;\;\;\;l\notin RL\left(b_{i}\right) \end{array}\right.$$

In the second method (REM2), the matrix completion technique is introduced to reduce the excessive culling of information. To be specific, the remaining SWT is first defined as ${{T_{x}^{remain}=T}_{x}-T}_{x}^{REM1}$. Then, the Singular Value Thresholding algorithm is used on $T_{x}^{remain}$ to generate a completed matrix $T_{x}^{com}$. Specifically, the Randomized Singular Value Decomposition (RSVD) is used to reduce the calculation time. Finally, the PLI signal is estimated by ISWT of:(15)$$T_{x}^{REM2}\left(\tilde{\xi_{l}},b_{i}\right)=\left\{\begin{array}{l} T_{x}\left(\tilde{\xi_{l}},b_{i}\right)-T_{x}^{com}\left(\tilde{\xi_{l}},b_{i}\right),\;\;\;l\in RL\left(b_{i}\right)\\ 0,\;\;\;\;\;\;\;\;\;\;\;\;l\notin RL\left(b_{i}\right) \end{array}\right.$$

## 3. Method Description

### 3.1. Simulation of the sEMG and PLI Signals

To evaluate the performance of different PLI filters, simulated sEMG and PLI signals were first generated. In this way, the synthetic signals with definite SNR and time–frequency characteristics can be obtained, which is beneficial to evaluate the performance under certain extreme conditions. The sEMG signal was implemented according to the following spectrum model [48]:(16)$$P\left(f\right)=\frac{f_{h}^{4}f^{2}}{\left(f^{2}{+f}_{l}^{2}\right){\left(f^{2}{+f}_{h}^{2}\right)}^{2}}$$

The sampling frequency f was set as 2 kHz. The frequency control parameters $f_{h}$ and $f_{l}$ were modified to simulate the process from excitation to fatigue. To be specific, $f_{h}$/$f_{l}$ gradually increased from 175/45 Hz to 200/60 Hz and then decreased to 150/30 Hz in 50 steps, with a random adjustment of the standard normal distribution being added to the frequency control parameters at each step. For each step, a 20-order recursive IIR filter with the frequency response described in Formula (16) was generated by the least square method. Then, a random white noise sequence of unit variance with 256 sampling points was processed through an IIR filter. After the 50 segments of processed signals were spliced and synthesized in sequence, amplitude modulation of the synthesized signal was performed. The modulated signals were the sEMG envelope generated from the tibialis anterior of the right leg of a 30-year-old healthy male during straight walking.

The PLI signals were simulated as: (17)$$PLI\left(t\right)=v_{srn}\left|\sin\left(2\pi tF_{s}/N_{sim}+\varphi_{rand}\right)\right|⨀\cos\left(f\left(t\right)\cdot 2\pi t+\theta \right)$$
where $N_{sim}$ refers to the length of the simulated signal. $v_{srn}\left|\sin\left(2\pi tF_{s}/N_{sim}+\varphi_{rand}\right)\right|$ is the amplitude-modulated signal, where $v_{srn}$ is determined by the input SNR. $f\left(t\right)$ is a sine function with $N_{sim}$ as its period, 1 Hz as its peak, and the central frequency as its mean. The central frequency is a random value between 49.8 Hz and 50.2 Hz. In addition, θ is set to be a random value between −π to π.

### 3.2. Obtaining of Real sEMG Signals

To analyze the performance of the filters processed with real sEMG signals, sampling from 32 patients with Wilson Disease and a healthy control group were recorded. The sEMG data of the lower extremity muscle were recorded for each subject as they performed lying, standing, sitting, the timed up-and-go test, and the ankle joint motion test in a sitting position. SEMG signals were recorded by NORAXON Ultium wireless sensors (Noraxon, Scottsdale, AZ, USA) at 2000 Hz. The Ultium sensors, which are reference electrodes, were placed on the skin over osseous structures or tendons. Disposable, self-adhesive Ag/AgCL dual electrodes, which are pairs of electrodes with a 2 cm center distance, were placed on the skin over the belly of the target muscles along the major orientation of muscle fibers. The skin of the participants over the target muscles was cleaned with an alcohol-soaked pad before setting up the sensors and electrodes. The target muscles were rectus femoris (near the midline of the thigh, approximately halfway between the ASIS and the proximal patella), semitendinosus (on the medial aspect of the thigh, located approximately 3 cm from the lateral border of the thigh and approximately half the distance from the hip to the back of the knee), tibialis anterior (lateral to the medial shaft of the tibia, at approximately one-third the distance between the knee and the ankle), lateral gastrocnemius (lateral to the back of shank, at approximately half the distance between the back of knee and heel), medial gastrocnemius (medial to the bank of shank, at approximately half the distance between the back of knee and heel), and peroneus longus (two-thirds of the way from the medial lateral center of the tibia to the distal end of the medial malleolus line) on both sides. Placement of the sensors and electrodes was based on the recommendation of the Surface EMG for Noninvasive Assessment of Muscles (SENIAM) [49].

Due to the influence of the differences between the subjects, experimental environment, execution level of the operation, and emergencies during the process, all the signals are more or less affected by the PLI. A rough index, which is the ratio of the sum of the spectrum energy in a small range near the estimated center frequency of PLI to the sum of the spectrum energy of the signal, was introduced to calculate the proportion of PLI noise in the real sEMG signal. Based on this, the most and least serious were selected for study. Considering that the pure sEMG signal in the real signal is unknown, it is not easy to select a suitable quantitative index to judge the performance of the filters. Therefore, the time–frequency spectrum diagram was used to check the performance qualitatively.

### 3.3. Performance Evaluation Criteria

In the evaluation of the performance of PLI-reducing methods in simulated signals, different noise intensities are introduced. First, the input SNR is introduced to evaluate the quality of the signal to be filtered, that is, the larger the input SNR, the less PLI in the signal. The input SNR is defined as:(18)$${SNR}_{in}=10\log_{10}\frac{\sum_{i=1}^{N_{sim}}{x\left(i\right)}^{2}}{\sum_{i=1}^{N_{sim}}{n\left(i\right)}^{2}}$$

Then, three performance indexes are introduced to evaluate the filtering performance of the PLI filters; namely, the output SNR:(19)$${SNR}_{out}=10\log_{10}\frac{\sum_{i=1}^{N_{sim}}{x\left(i\right)}^{2}}{\sum_{i=1}^{N_{sim}}{\left(x\left(i\right)-x_{d}\left(i\right)\right)}^{2}}$$
the correlation coefficients between the denoised signal and the raw sEMG signal:(20)$$CC=\frac{\sum_{i=1}^{N_{sim}}x\left(i\right)x_{d}\left(i\right)}{\sqrt{\sum_{i=1}^{N_{sim}}{x\left(i\right)}^{2}\sum_{i=1}^{N_{sim}}{x_{d}\left(i\right)}^{2}}}$$
and the output SNR and the root mean square error (RMSE):(21)$$RMSE=\sqrt{\frac{\sum_{i=1}^{N_{sim}}{\left(x\left(i\right)-x_{d}\left(i\right)\right)}^{2}}{N_{sim}}}$$
where $x\left(i\right)$, $n\left(i\right)$, and $x_{d}\left(i\right)$ denote the raw sEMG signal, the PLI signal, and the denoised signal, respectively.

### 3.4. Optimization of the Parameters

The mother wavelet is expected to have a small frequency support range, which leads to more accurate ridge localization. In the present paper, the bump wavelet was selected [38,39], which has the Fourier transform:(22)$$\hat{\varphi }\left(\xi \right)={exp}^{1-\frac{1}{1-{\left(\frac{\xi -\mu }{\sigma }\right)}^{2}}}\chi_{\left[\mu -\sigma ,\mu +\sigma \right]}$$
where $\chi_{\left[\mu -\sigma ,\mu +\sigma \right]}$ is an indicator function of the set $\left[\mu -\sigma ,\mu +\sigma \right]$. The bump wavelet admits a unique peak frequency ξ=μ and is supported in the range of $\left[\mu -\sigma ,\mu +\sigma \right]$. By optimizing the parameters μ and σ through the experiment, the energy aggregation of the parent wavelet in the time domain and frequency domain can be adjusted.

Thirty simulated sEMG signals with PLI noise were generated randomly. To be specific, the preset filtering bandwidth (${2\xi }_{w}$) and the estimated center frequency ($\xi_{c}$) of PLI noise were set to 6 Hz and 50 Hz, respectively. The input SNRs of PLI were set to be −20, −10, 0, 10, and 20 dB. Three kinds of parameter were optimized, which were μ, σ, and the resolution $\xi_{r}$; the value ranges of the parameters are shown in Table 1. The choice of the parameters was determined by comparing the mean normalized output SNR and computation time after running the proposed algorithm with REM1 and REM2 in the 30 simulated signals. The mean normalized output SNR is defined as:(23)$${SNR}_{out\_norm}=\frac{1}{30}\sum_{i=1}^{30}\frac{{SNR}_{out}^{i}-{SNR}_{out}^{min}}{{SNR}_{out}^{max}-{SNR}_{out}^{min}}$$
where ${SNR}_{out}^{i}$, ${SNR}_{out}^{min}$, and ${SNR}_{out}^{max}$ are the output SNR of the i-th simulated signals, and the minimum and maximum output SNR of all the simulated signals, respectively.

### 3.5. Statistic Analysis

A total of 314 synthesized signals were randomly generated for each input SNR (−20 dB, −10 dB, 0 dB, 10 dB, and 20 dB). The proposed methods (REM1 and REM2), the notch filtering, and spectral interpolation were performed on each synthesized signal. Specifically, liner interpolation was used in spectral interpolation, and two different bandwidths of 1 Hz and 6 Hz were used for notch filtering. The Friedman test and Wilcoxon signed-rank test were used to evaluate the difference in the performance among different methods under various input conditions. In addition, the proposed method and other filters were performed on the real sEMG signal as described in Section 3.2.

## 4. Results

Results are shown in this section. First, the filtering performance and computing time of the proposed methods with parameters described in Section 3.4 are presented in Section 4.1. Then, setting the parameters to the most appropriate value, the quantitative indexes described in Section 3.3 are shown in Section 4.2 after processing the simulated signals by the proposed and compared methods, and the qualitative evaluation results are given after processing the real signals in Section 4.3.

### 4.1. Parameters of the LSWT

The performance of the proposed algorithms used with different parameters is shown in Figure 3. Basically, in Figure 3a,b, the normalized output SNR increases as μ increases or σ decreases for both REM1 and REM2, although there is an obvious anomaly when $\xi_{r}$ equals 1. By comparison, when $\xi_{r}$ changes, the normalized output SNR does not change approximately monotonously. To be specific, the normalized output SNR when $\xi_{r}$ equals 0.2, 0.33, or 0.5 is significantly greater than that when $\xi_{r}$ equals 0.1, 0.75, or 1. The largest three normalized output SNR values are achieved when (μ, σ, $\xi_{r}$) are equal to (8, 0.2, 0.33), (8, 0.2, 0.5), and (6, 0.2, 0.33) in descending order when using REM1, while they are achieved when (μ, σ, $\xi_{r}$) are equal to (7, 0.2, 0.33), (8, 0.2, 0.33), and (8, 0.2, 0.5) when using REM2. Meanwhile, in Figure 3c,d, the computation time decreases as μ increases, σ decreases, or $\xi_{r}$ increases for both REM1 and REM2.

The most appropriate parameters were selected and all the following results were calculated using these selected parameters.

### 4.2. Performance of the PLI Filters in Simulated Signals

The distributions of the output SNR of the proposed methods and the other filters under different input SNR values are shown in Figure 4. Except for the notch filtering with wide bandwidth, the output SNR of each method increases significantly with the increase in the input SNR. For each input SNR, the significance probabilities of Wilcoxon signed-rank tests on the output SNR of almost every pair of methods are below 0.05; the only exception is the output SNR of REM1 and REM2 when the input SNR is −20 dB. In addition, the performances of REM1 and REM2 are significantly higher those of the other methods. Excluding notch filtering with wide bandwidth, the methods’ ascending order in terms of the output SNR for each input condition are notch filtering with narrow bandwidth, spectral interpolation, REM1, and REM2.

Table 2, Table 3 and Table 4 reveal the mean value ± standard deviation of the output SNR, CC, and RMSE obtained from each filtering method, classified according to the input SNR. The notch filtering with narrow bandwidth and spectral interpolation presents a bad performance when the input SNR is low, but this is improved when the input SNR is increased. Notch filtering with wide bandwidth shows a good performance at low input SNR, but its performance hardly improves with the increase in the input SNR. Even worse, the performance of notch filtering with wide bandwidth after processing is worse than that before processing in the case of a high input SNR (20 dB). In contrast, the proposed methods provide the best performance at both low and high input SNR values. Meanwhile, the performance of REM2 is better than that of REM1 at a high input SNR, and the performances of the two are nearly equal at a low input SNR. To be specific, in the case of the low input SNR (−20 dB), the output SNR of the proposed filters (REM1/RME2) is 28.51/28.55, 23.24/23.28, and 1.64/1.6 higher than that of notch filtering with narrow bandwidth, spectral interpolation, and notch filtering with wide bandwidth, respectively. When the input SNR is high (20 dB), the output SNR of the proposed methods is 3.33/5.68, 2.07/4.42, and 7.4/9.75 higher than that of the other methods mentioned above, respectively. Meanwhile, the RMSE values of the above three commonly used filters are 25.1/25.7, 13.6/13.9, and 1.12/1.14 times those of RME1/REM2 at a low input SNR, while they are 1.71/2.42, 1.47/2.08, and 2.71/3.83 times those of REM1/REM2 at a high input SNR.

### 4.3. Performance of the PLI Filters in Real Signals

The time–frequency spectra of the raw and processed sEMG signals obtained from the two subjects whose sEMG was most and least affected by PLI are shown in Figure 5 and Figure 6, respectively. In Figure 5a and Figure 6a, bright lines can be observed along the time axis near the power line frequency, which represents the time–frequency spectral energy of the PLI. The goal of the PLI filtering is to eliminate the bright lines mentioned above while preserving the background image, which represents the time–frequency spectral energy of the sEMG signal. Figure 5b,c and Figure 6b,c show that when processing the real sEMG signals, whether using notch filtering with narrow or wide bandwidth, PLI is eliminated but useful information is also removed, which is manifested as the obvious light-colored regions near the power line frequency in the figures. Figure 5d and Figure 6d reveal that, unlike the notch filtering, spectral interpolation suppresses rather than eliminates the PLI, but this suppression is not reliable in real signals. For instance, in Figure 5d, the bright lines representing the PLI are significantly weakened, but in Figure 6d they are somewhat enhanced. Compared with other methods, the proposed filter achieves the goal better; as shown in Figure 5e,f and Figure 6e,f, the bright lines representing the PLI are removed while the background images are well preserved.

In particular, a strong disturbance was observed at about 25 s from the time–frequency spectrum diagram of the first subject. In fact, the PLI is so severe that the spot at that moment overwhelms the image elsewhere in the diagram. To solve this problem, the logarithm of the spectrum energy is introduced to reduce the difference in scale. Figure 5 and Figure 6 show the results of this process. The abovementioned signal, which is heavily affected by PLI, is shown in Figure 7a, while Figure 7b–f show the results of the signal processed by the compared and proposed filters. In addition, the spectra between 47 and 54 Hz of these signals are shown if Figure 8.

All the PLI filters were implemented in MATLAB R2018a, and the statistical processes were performed in SPSS 24 on a 1.8 GHz Inter(R) Core(TM) i5 processor with 8 GB RAM.

## 5. Discussion and Conclusions

In sEMG signals, PLI is a major source of noise whose spectrum overlaps with useful signals. It seriously affects the reasonable interpretation of sEMG signals. As shown in Figure 1b, obvious morphological differences exist between sEMG and PLI signals on the time–frequency spectrum diagram. The former is discrete on the time axis and approximately continuous on the frequency axis, while the latter has the opposite pattern and is clustered around the power line frequency and its higher harmonics. Based on this, the problem of PLI removal is transformed into the problem of ridge extraction in the local time–frequency spectrum. In fact, the advantage of using the ridge extraction technique for noise removal from physiological signals has been demonstrated in studies [38,40], but no example of its application to PLI filtering on sEMG has been found. As sEMG can be easily affected by the noise, and it is easy to introduce strong time-varying PLI in extreme cases, it is difficult to ensure the ability of signal reconstruction and appropriate time–frequency energy concentration at the same time by using conventional time–frequency processing methods. As a countermeasure, the synchrosqueezed wavelet transform is introduced in this study. The present study proposes an PLI filter for the sEMG signal based on the SWT. After emphasizing the serious waste of computing time and space in calculating the global wavelet scale, the local CWT/SWT used to calculate the scale related to the expected PLI frequency range were developed, and then the parameters used to adjust the local resolution were given. 

Figure 3 shows the relations between the values of the parameters and the performance of the proposed filters. According to Formula (8), the wavelet scale control factor $n_{v}$ is inversely proportional to the resolution, so an appropriate increase in resolution can greatly reduce the calculation cost, which is consistent with Figure 3c,d. On the contrary, the resolution cannot be increased indefinitely, which leads to insufficient time–frequency energy concentration and thus reduces the performance of the filter. According to Figure 3, $\xi_{r}$ = 0.5 is a suitable choice: the output SNR decreases significantly when $\xi_{r}$ increases to 0.75, while the improvement in output SNR achieved by reducing $\xi_{r}$ to 0.33 is not high. In particular, in Figure 3a,b, $\xi_{r}$ = 0.5 is found in the three best parameter sets. In addition, filtering performance and computing cost are also affected by μ and σ. This is obvious because, as shown in Formula (22), the time–frequency resolution of the mother wavelet can be adjusted by changing these two parameters, thus affecting the aggregation of the time–frequency spectrum energy after the SWT. In addition, μ and σ also affect the calculation of the smallest cardinality in Formula (7), thus affecting the calculation cost, which can be seen in Figure 3c,d. In summary, *(*μ, σ, $\xi_{r}$) is finally set to (8, 0.2, 0.5).

Based on the information of the neighborhood bandwidth defined in Section 2.3 and Section 2.5, an adaptive threshold parameter and ridge location method are introduced, so that the time-varying PLI ridges in a wide bandwidth (defined as 6 Hz) can be captured adaptively. Two ridge extraction methods are described: REM1 is used to remove the signal obtained by the inverse transform of the ridge time–frequency information directly from the raw sEMG signal, and REM2 uses the matrix completion method to modify the former. Comparing Figure 3a and Figure 3b, the normalized output SNRs obtained by REM2 are generally higher than those obtained by REM1, which reveals that REM2 is more stable than REM1 in processing time-varying signals at given parameter values. The possible reason for this is that the additional matrix completion in REM2 makes the information around the PLI ridges more consistent with its neighborhood information, which reduces the risk of over-eliminating useful information in extreme cases. In addition to stability, this advantage also leads to higher filtering performance. As Figure 4 and Table 2, Table 3 and Table 4 reveal, the output SNRs obtained by REM2 are higher than those obtained by REM1 in the case where the input SNR equals −10, 0, 10, and 20, while they are equal in the case where the input SNR is set to −20. Comparing different input conditions, an increase in the advantage of REM2 over REM1 on the output SNR is observed with the increase in the input SNR. This is because when the proportion of PLI is low, the retention of useful signals becomes more critical. On the contrary, as the iterations of matrix completion in REM2 require too many steps of SVD, the computing cost increases sharply compared with REM1. To blunt this impact, RSVD is used to replace SVD. However, as shown in Figure 3c,d, REM2 is still an order of magnitude slower than REM1. Therefore, REM1 is suitable for online filtering with high computing speed, while REM2 is more suitable for offline filtering with abundant computing resources.

In Figure 4 and Table 2, Table 3 and Table 4, except for notch filtering with wide bandwidth, the performance of the filters improves with the increase in input SNR in processing simulated signals. The possible reason for this is that the notch filtering with wide bandwidth removes both PLI and useful information in a wide bandwidth. As a result, the signal quality after filtering is unexpectedly lower than before processing when the input SNR is high. In Figure 5c and Figure 6c, obvious pits near the power line frequency can be observed during the whole period, indicating that notch filtering with wide bandwidth also has the above defects in real EMG processing. By contrast, notch filtering with narrow bandwidth can reduce the elimination of useful information, but cannot remove the PLI effectively when the input SNR is low. In the processing of real signals, both methods lead to a significant loss of useful information during muscle activation. When dealing with the synthetic signals, the performance of spectral interpolation is better than that of notch filtering with narrow bandwidth; this finding is similar to the result described in [30]. However, it can improve when dealing with long time real signals. The possible reason for this is that PLI is modeled as a time-invariant signal in spectral interpolation, while the real signal is time-varying, and the increase in signal length amplifies the problem caused by this contradiction. A feasible improvement method is to use a sliding window to divide the long time series into several short sequences, and process each sequence sequentially before synthesizing it. This method is similar to STFT. However, it still cannot deal with the problem of poor time–frequency energy aggregation and cannot adaptively adjust the interpolation bandwidth. Compared with the above filters, the proposed method introduces SWT to process time-varying PLI signals, and uses the adaptive ridge extraction method to balance noise filtering and useful signal retention. Thus, the best evaluation indexes were obtained when processing simulated signals using the proposed filters, and the optimal time–frequency spectrum diagrams are displayed when processing real signals. 

Furthermore, Figure 7 and Figure 8 demonstrate the performance of different filters in dealing with extreme cases. Figure 7a presents a piece of real sEMG signal doped with large-amplitude time-varying PLI. Figure 7b,d shows that PLI filtering with narrow bandwidth or spectral interpolation is incomplete when dealing with extreme cases. This is evidenced by the high spectral values retained near the power frequency in Figure 8a. In Figure 7c,e,f, signals obtained using notch filtering with wide bandwidth, REM1, and REM2 have no obvious image differences in the time domain. On the contrary, obvious differences can be observed in the frequency domain. In Figure 8b, it can be observed that the spectral value near 50Hz is almost zero for notch filtering with wide bandwidth, while a gentle curve of continuous change is preserved in this region for REM1 and REM2. This shows that, even in real, extreme cases, the proposed filters can both filter out sufficient PLI and reduce the undesirable elimination of useful information.

In summary, the proposed algorithms presented significantly better performance in both simulated and real sEMG signal processing.

## Figures and Tables

**Figure 1 sensors-23-05182-f001:**
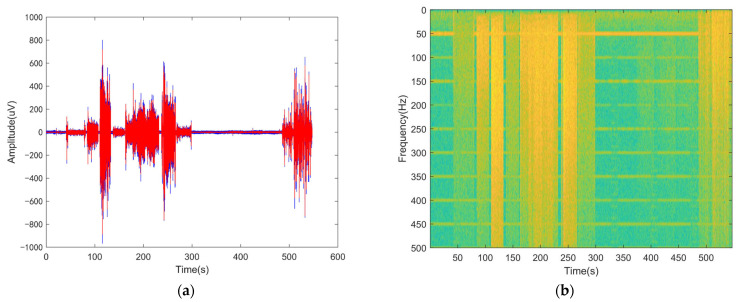
(**a**) Typical signal of sEMG mixed with PLI, where the blue line is the raw signal and the red line represents the signal processed by a notch filter. (**b**) The time-spectrum diagram obtained using a short-time Fourier transform.

**Figure 2 sensors-23-05182-f002:**
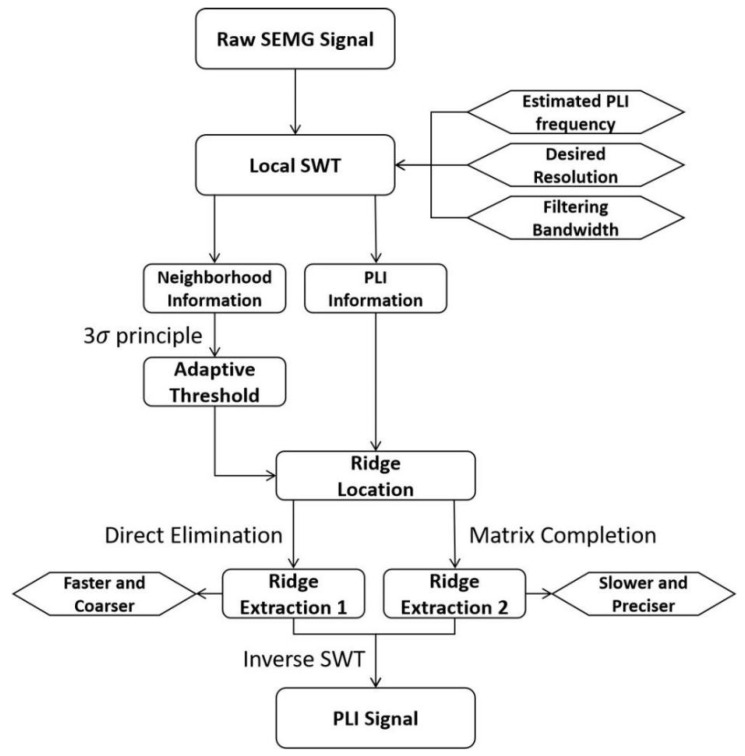
Flowchart of the SWT-based PLI filtering method for the sEMG signal.

**Figure 3 sensors-23-05182-f003:**
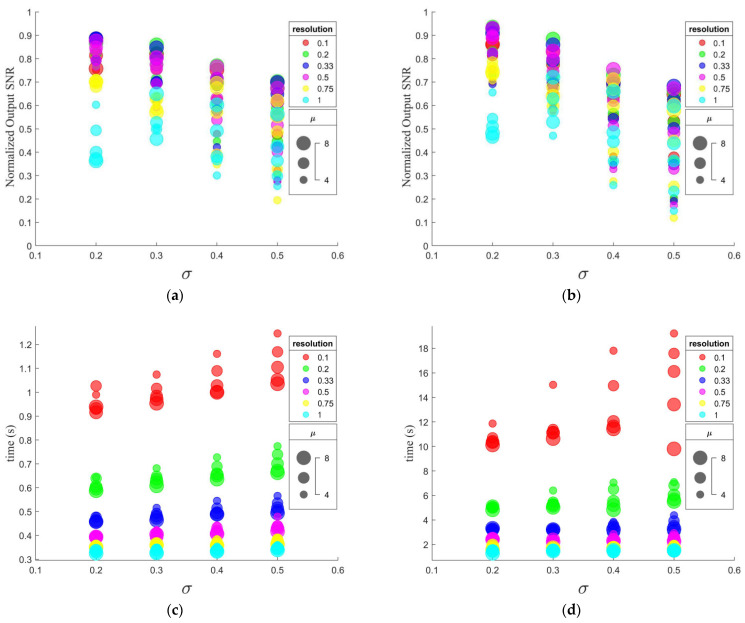
Bubble diagram showing the performances of the proposed filters used with different parameters: (**a**) mean normalized output SNR of REM1; (**b**) mean normalized output SNR of REM2; (**c**) computation time of REM1; (**d**) computation time of REM2.

**Figure 4 sensors-23-05182-f004:**
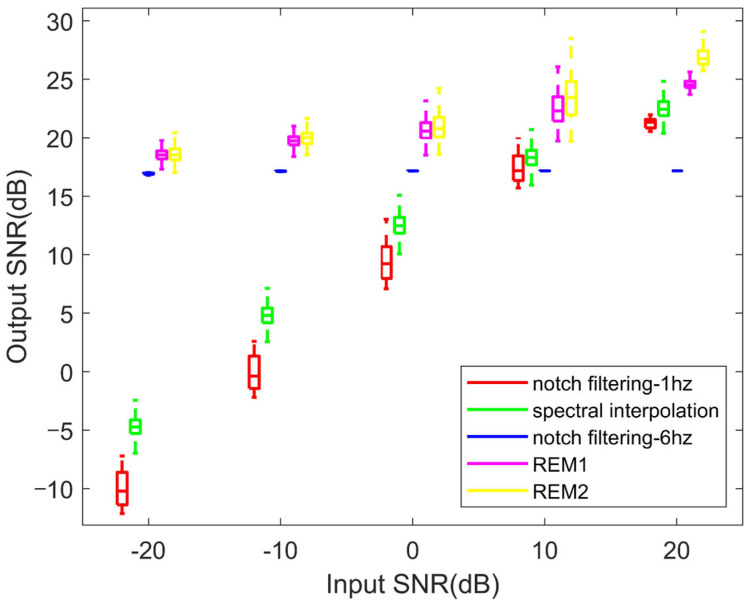
A comparison of the output SNR from five different PLI filtering methods under different input SNR values.

**Figure 5 sensors-23-05182-f005:**
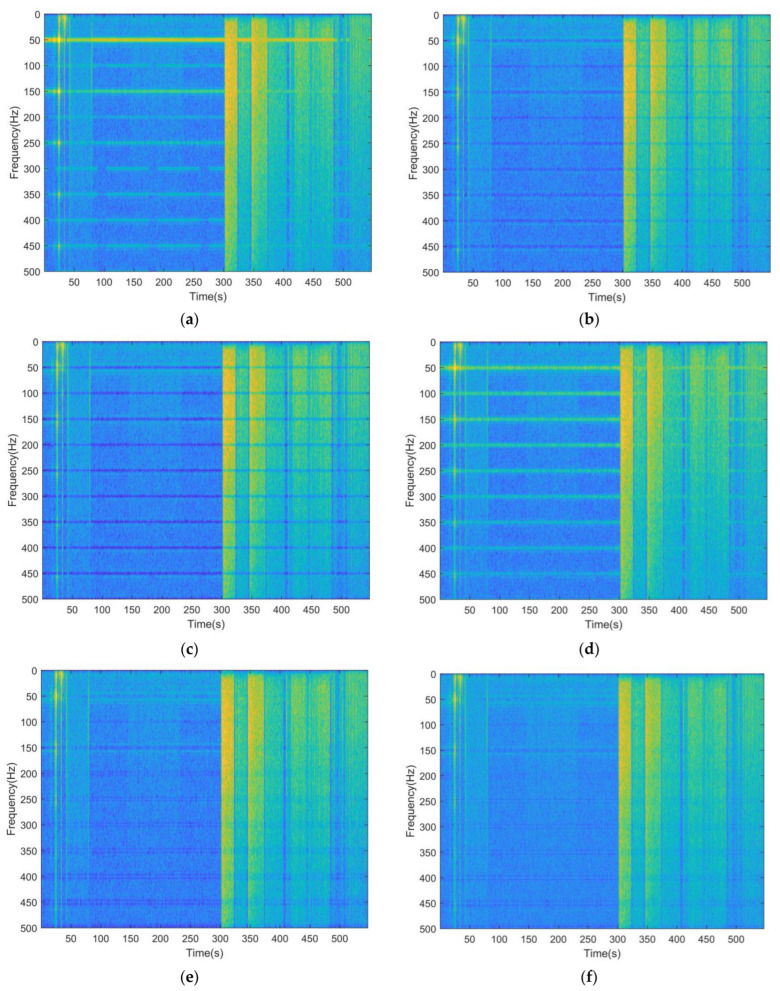
Time–frequency spectrum diagram of the raw and processed signals of the subject whose sEMG signal is most affected by PLI: (**a**) raw sEMG; (**b**) signal processed by the notch filter (narrow bandwidth); (**c**) signal processed by the notch filter (wide bandwidth); (**d**) signal processed by spectral interpolation; (**e**) signal processed by REM1; (**f**) signal processed by REM2.

**Figure 6 sensors-23-05182-f006:**
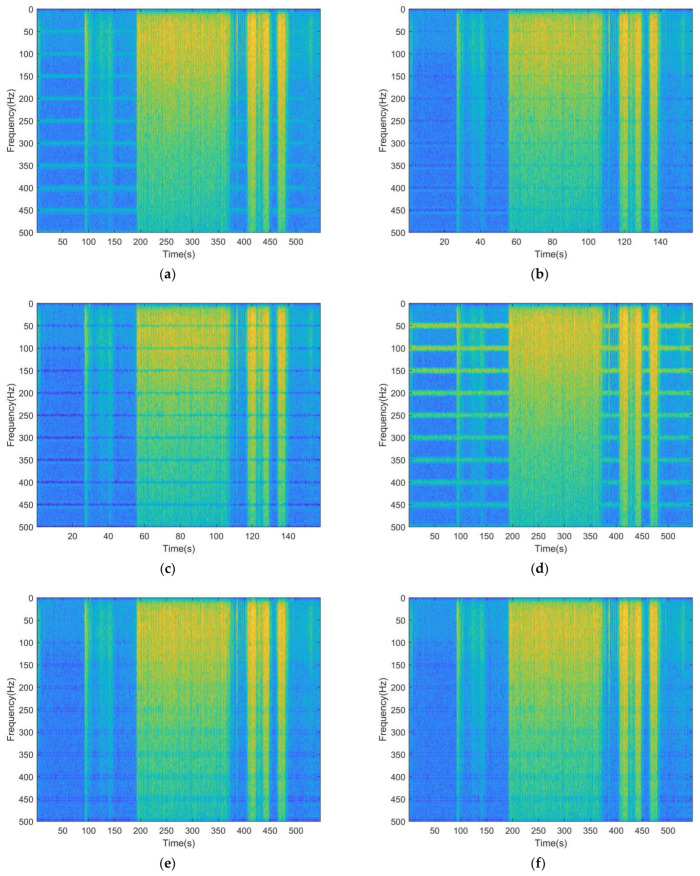
Time–frequency spectrum diagram of the raw and processed signals of the subject whose sEMG signal is least affected by PLI: (**a**) raw sEMG; (**b**) signal processed by the notch filter (narrow bandwidth); (**c**) signal processed by the notch filter (wide bandwidth); (**d**) signal processed by spectral interpolation; (**e**) signal processed by REM1; (**f**) signal processed by REM2.

**Figure 7 sensors-23-05182-f007:**
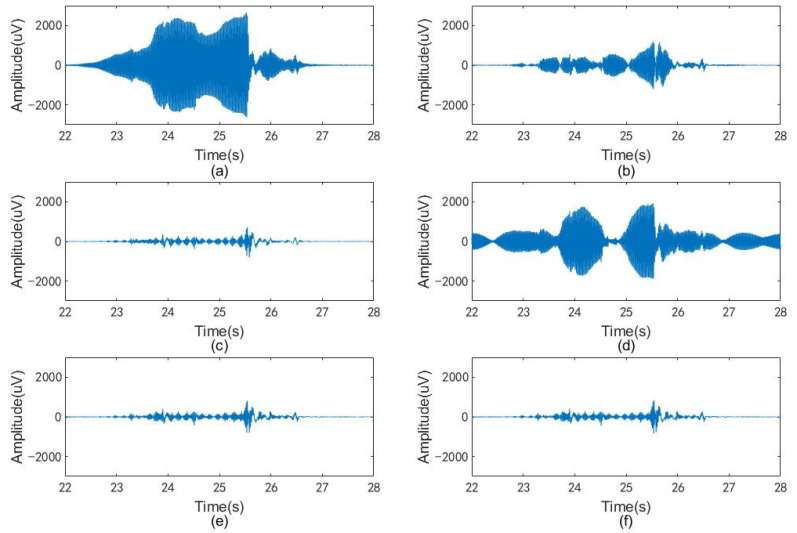
A section of the real sEMG signal that is heavily affected by PLI after processing by different algorithms: (**a**) raw sEMG; (**b**) signal processed by the notch filter (narrow bandwidth); (**c**) signal processed by the notch filter (wide bandwidth); (**d**) signal processed by spectral interpolation; (**e**) signal processed by REM1; (**f**) signal processed by REM2.

**Figure 8 sensors-23-05182-f008:**
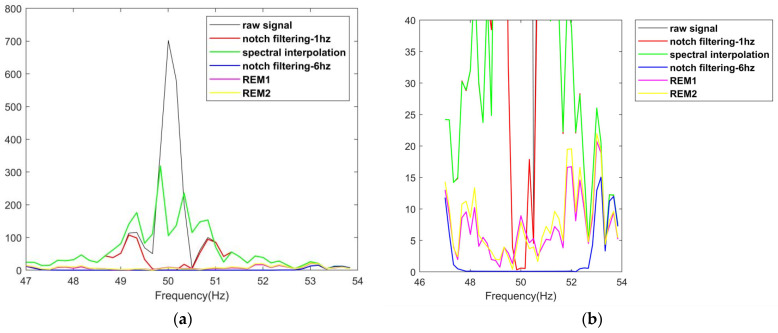
The spectra (47–54 Hz) of the raw and processed signals in Figure 7: (**a**) general view; (**b**) local plot of lower value.

**Table 1 sensors-23-05182-t001:** The values of parameters used in the parameter-optimizing experiment.

Parameter	The Value Range of the Parameter
μ	[4, 5, 6, 7, 8]
σ	[0.2, 0.3, 0.4, 0.5]
$\xi_{r}$	[0.1, 0.2, 0.33, 0.5, 0.75, 1]

**Table 2 sensors-23-05182-t002:** Output SNR of the processed signals (mean ± standard deviation from 314 random simulated signals).

Input SNR of the Simulated Signal	−20 dB	−10 dB	0 dB	10 dB	20 dB
Notch filtering with 1 Hz bandwidth [26,27,28]	−9.98 ± 1.54	−0.13 ± 1.48	9.43 ± 1.61	17.42 ± 1.20	21.24 ± 0.39
Spectral interpolation [27,30]	−4.71 ± 0.93	4.85 ± 0.95	12.53 ± 0.98	18.37 ± 0.95	22.50 ± 0.95
Notch filtering with 6 Hz bandwidth [26,27,28]	16.93 ± 0.56	17.14 ± 0.02	17.16 ± 0.05	17.17 ± 0.02	17.17 ± 0.00
REM1	18.53 ± 0.49	19.74 ± 0.52	20.62 ± 0.95	22.51 ± 1.40	24.57 ± 0.73
REM2	18.57 ± 0.64	19.98 ± 0.67	20.98 ± 1.30	23.47 ± 1.79	26.92 ± 0.77

**Table 3 sensors-23-05182-t003:** Correlation coefficients between the raw sEMG signals and the processed signals (mean ± standard deviation from 314 random simulated signals).

Input SNR of the Simulated Signal	−20 dB	−10 dB	0 dB	10 dB	20 dB
Without filtering	0.19 ± 0.02	0.53 ± 0.01	0.89 ± 0.00	0.97 ± 0.00	0.97 ± 0.00
Notch filtering with 1 Hz bandwidth [26,27,28]	0.29 ± 0.05	0.69 ± 0.06	0.93 ± 0.02	0.98 ± 0.00	0.99 ± 0.00
Spectral interpolation [27,30]	0.50 ± 0.04	0.87 ± 0.02	0.97 ± 0.01	0.99 ± 0.00	0.99 ± 0.00
Notch filtering with 6 Hz bandwidth [26,27,28]	0.95 ± 0.00	0.97 ± 0.00	0.98 ± 0.00	0.98 ± 0.00	0.98 ± 0.00
REM1	0.99 ± 0.00	0.99 ± 0.00	0.99 ± 0.00	1.00 ± 0.00	1.00 ± 0.00
REM2	0.99 ± 0.00	0.99 ± 0.00	0.99 ± 0.00	1.00 ± 0.00	1.00 ± 0.00

**Table 4 sensors-23-05182-t004:** Root mean square error of the processed signals (mean ± standard deviation from 314 random simulated signals).

Input SNR of the Simulated Signal	−20 dB	−10 dB	0 dB	10 dB	20 dB
Without filtering	3646 ± 1	1153 ± 0	365 ± 0	11 ± 0	54 ± 0
Notch filtering with 1 Hz bandwidth [26,27,28]	2131 ± 369	685 ± 113	229 ± 41	90 ± 12	58 ± 3
Spectral interpolation [27,30]	1152 ± 123	383 ± 41	158 ± 18	81 ± 9	50 ± 5
Notch filtering with 6 Hz bandwidth [26,27,28]	95 ± 1	93 ± 0	92 ± 0	92 ± 0	92 ± 0
REM1	85 ± 5	72 ± 4	60 ± 7	47 ± 8	34 ± 2
REM2	83 ± 6	70 ± 5	57 ± 9	41 ± 8	24 ± 3

## Data Availability

Data in this manuscript is not available to publish due to privacy restrictions.

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
