# Peer review of "Reducing Power Line Interference from sEMG Signals Based on Synchrosqueezed Wavelet Transform"

_sensors, 2023, doi:10.3390/s23115182_

Round 1

Reviewer 1 Report

The authors present the paper entitled “Reducing Power Line Interference from sEMG Signals Based  on Synchrosqueezed Wavelet Transform”

This paper proposes a novel adaptive PLI filter based on the synchrosqueezed wavelet transform (SWT) and proposes a  time-frequency spectrum ridge location method based on an adaptive threshold.

The article presents the following concerns:

  • Add hyperlinks to tables, figures, and references.

  • Avoid using personal pronouns like We and use the passive voice instead.

  • Reduce the percentage of plagiarism to 20% since it is over 24% according to turnitin.

  • Authors are asked to add in their abstract the most critical quantitative results.

  • Avoid using contractions

  • Add a short introduction between section 2 and subsection 2.1.

  • Authors are asked to add a flowchart that exemplifies the methodology

  • On line 217, correcting SEMG.

  • In section 3.2, it is necessary to mention under what standards the placement of the electrodes was carried out or what consideration was taken into account.

  • Line 228 mentions, "The most and least serious were selected for the study." Therefore, the selection criteria should be mentioned to avoid ambiguities.

  • Edit "signal. respectively." On line 238.

  • Add a short introduction between section 4 and subsection 4.1.

  • Improve the quality of Figures 2, 3, 4, and 5.

  • In Figure 2, correcting the writing of the decimals from .1 to 0.1 is presented in the resolution legend.

  • Sentence “…One obstacle is that the raw sEMG signal is susceptible to noise and may have a low signal-to-noise ratio (SNR)…” can be justified with the following references: A novel methodology for classifying emg movements based on svm and genetic algorithms; A study of computing zero crossing methods and an improved proposal for emg signals; Support vector machine-based emg signal classification techniques: a review; A study of movement classification of the lower limb based on up to 4-emg channels

  • It is necessary to analyze and explain each of the subfigures of Figure 2.

  • It is necessary to analyze and explain Figure 3.

  • Rearrange the figures and tables, and place them after they are mentioned.

  • Add a comparative table where the main results of this work are shown and compared with others to highlight the findings presented.

The following misspelling should be checked:

  1. line 52: The phrase “To be specific” may be wordy. Consider changing by “Specifically”

  2. line 105: “briefly” sounds redundant, eliminate it.

  3. line 219: “difficult” may sound overly negative to your reader. Consider rephrasing it: “isn’t easy”

  4. line 325: “In Figure 4(a) and 5(a)” It seems taht “figure” may not agree in number with other words in this phrase. Changing by “In Figures 4(a) and 5(a)”. same case in lines 329, 359.

  5. line 371: It appears that the form of the verb “exist” does not work with “are” in this sentence. Eliminate “are.”

  6. line 407: “But it does not do well when dealing with long time real signals…” should be rewritten by “But it could do better when dealing with long-time accurate signals….”

Author Response

Q1. Add hyperlinks to tables, figures, and references.

Hyperlinks have been added.

Q2. Avoid using personal pronouns like We and use the passive voice instead.

The modification have been done.

Q3. Reduce the percentage of plagiarism to 20% since it is over 24% according to turnitin.

It is 13% now.

Q4. Authors are asked to add in their abstract the most critical quantitative results.

The abstract have been modified, the output SNR of proposed methods have been added.

Q5. Avoid using contractions

The modification have been done.

Q6. Add a short introduction between section 2 and subsection 2.1.

A short introduction have been added at this location, and a flowchart of the proposed method have also been added in this introduction.

Q7. Authors are asked to add a flowchart that exemplifies the methodology

A new figure (Figure 2) have been added to show the flowchart.

Q8. On line 217, correcting SEMG.

The modification have been done.

Q9. In section 3.2, it is necessary to mention under what standards the placement of the electrodes was carried out or what consideration was taken into account.

The detailed description of the electrodes placement have been added in section 3.2. And a declaration of the criteria to be referenced have been added: ‘Placement of the sensors and electrodes were based on the recommendation of the Surface EMG for Noninvasive Assessment of Muscles (SENIAM)’.

Q10. Line 228 mentions, "The most and least serious were selected for the study." Therefore, the selection criteria should be mentioned to avoid ambiguities.

A supplementary statement on this question have been added :

‘A rough index, which is the ratio of the sum of the spectrum energy in a small range near the estimated centre frequency of PLI to the sum of the spectrum energy of the signal, was introduced to calculate the proportion of PLI noise in the real SEMG signal. Based on this, the most and least serious ones were selected to study’

Q11. Edit "signal. respectively." On line 238.

The modification have been done.

Q12. Add a short introduction between section 4 and subsection 4.1.

A paragraph have been added in this location.

Q13. Improve the quality of Figures 2, 3, 4, and 5.

Reediting and typesetting have been done for these figures, and Figures 7 and 8 were added to better illustrate the performance of the proposed and compared filters in processing real signals.

Q14. In Figure 2, correcting the writing of the decimals from .1 to 0.1 is presented in the resolution legend.

The modification have been done.

Q15. Sentence “…One obstacle is that the raw sEMG signal is susceptible to noise and may have a low signal-to-noise ratio (SNR)…” can be justified with the following references: A novel methodology for classifying emg movements based on svm and genetic algorithms; A study of computing zero crossing methods and an improved proposal for emg signals; Support vector machine-based emg signal classification techniques: a review; A study of movement classification of the lower limb based on up to 4-emg channels

The modification have been done.

Q16. It is necessary to analyze and explain each of the subfigures of Figure 2.

Analysis and explanation of Figure2 (now Figure 3) have been added and adjusted in section 5.

Q17. It is necessary to analyze and explain Figure 3.

Analysis and explanation of Figure3 (now Figure 4) have been added and adjusted in section 4.2 and 5.

Q18. Rearrange the figures and tables, and place them after they are mentioned.

The modification have been done.

Q19. Add a comparative table where the main results of this work are shown and compared with others to highlight the findings presented.

Tabel 2-4 shows the main results of this work are shown and compared with others, citation have been added.

Q20. line 52: The phrase “To be specific” may be wordy. Consider changing by “Specifically”

The modification have been done.

Q21. line 105: “briefly” sounds redundant, eliminate it.

The modification have been done.

Q22. line 219: “difficult” may sound overly negative to your reader. Consider rephrasing it: “isn’t easy”

The modification have been done.

Q23. line 325: “In Figure 4(a) and 5(a)” It seems that “figure” may not agree in number with other words in this phrase. Changing by “In Figures 4(a) and 5(a)”. same case in lines 329, 359.

The modification have been done.

Q24. line 371: It appears that the form of the verb “exist” does not work with “are” in this sentence. Eliminate “are.”

The modification have been done.

Q25. line 407: “But it does not do well when dealing with long time real signals…” should be rewritten by “But it could do better when dealing with long-time accurate signals….”

The modification have been done.

Reviewer 2 Report

The following aspects should be adressed:

The modulated signal were the sEMG -> signals were the SEMG

Obtain of Real sEMG signals - > obtaining

data of lower extremity muscle were -> was

skin over the target muscles was cleaning -> was cleaned

The target muscles are  -> were 

Due to the influence of the difference -> differences

In case of evaluating the performance of PLI reducing methods in simulated signals, different noise intensities are introduced, which are the input SNR, the correlation coefficients between the denoised signal and the raw sEMG signal, the output SNR and the root mean square error (RMSE) , as: - > unclear, please rewrite

mother wavelet is expected to has -> have

The bump wavelet admit -> admits

To be specific, the preset filtering bandwidth, center frequency of PLI noise are 6 Hz and 50 Hz, respectively.  - unclear

Lines 269 to 274 seem to be spaced differently from the rest of the text. Are the special characters' fonts correct?

I recommend splitting figures 4 and 5 into multiple separate plots, and placing them in a way that allows for easy comparison between cases. It is hard to compare, for example, Figure 4(b), 4(c), 5(b) and 5(c) (as mentioned in line 329) when it requires skipping between pages. 

In particular, a strong disturbance was observed at about 25 seconds from the first subject's time-frequency spectrum diagram. In fact, the PLI is so severe that the spot at that moment overwhelms the image elsewhere in the diagram - > can You show that disturbance in a zoomed in plot before and after filtering? These are always hard to process and the reader might be curious how the methods peformed in this extreme case. 

In fact, could You provide a few plots with interesting cases before and after filtering using Your and other methods? Seeing SNR numbers and spectrum is okay, but this will certainly improve the quality of Your presentation.

Discussion and conclusions paragraph has different fonts from line 362 to 370.

You should also mention other methods of signal analysis, like EMD and EEMD, which can be applied to non-linear and non-stationary signals. Please consider adding the following sources DOI 10.3390/s22103765, DOI 10.35784/acs-2022-14, DOI 10.3390/s22062176 in the introduction section. 

Comments were presented in the section before.

Author Response

Q1. The modulated signal were the sEMG -> signals were the SEMG

The modification have been done.

Q2. Obtain of Real sEMG signals - > obtaining

The modification have been done.

Q3. data of lower extremity muscle were -> was

The modification have been done.

Q4. skin over the target muscles was cleaning -> was cleaned

The modification have been done.

Q5. The target muscles are  -> were

The modification have been done.

Q6. Due to the influence of the difference -> differences

The modification have been done.

Q7. In case of evaluating the performance of PLI reducing methods in simulated signals, different noise intensities are introduced, which are the input SNR, the correlation coefficients between the denoised signal and the raw sEMG signal, the output SNR and the root mean square error (RMSE) , as: - > unclear, please rewrite

This paragraph have been modified as:
In case of evaluating the performance of PLI reducing methods in simulated signals, different noise intensities are introduced. First, the input SNR is introduced to evaluate the quality of the signal to be filtered, that is, the larger the input SNR is, the less PLI is in the signal. The input SNR is defined as:

(18)

Then, three performance indexes are introduced to evaluate the filtering performance of the PLI filters. Namely, the output SNR:

(19)

the correlation coefficients between the denoised signal and the raw SEMG signal:

(20)

and the output SNR and the root mean square error (RMSE) :

(21)

Q8. mother wavelet is expected to has -> have

The modification have been done.

Q9. The bump wavelet admit -> admits

The modification have been done.

Q10. To be specific, the preset filtering bandwidth, center frequency of PLI noise are 6 Hz and 50 Hz, respectively.  - unclear

This sentence have been modified to ‘ To be specific, the preset filtering bandwidth () and the estimated center frequency () of PLI noise are set to 6 Hz and 50 Hz, respectively. ’.

Q11. Lines 269 to 274 seem to be spaced differently from the rest of the text. Are the special characters' fonts correct?

The modification have been done.

Q12. I recommend splitting figures 4 and 5 into multiple separate plots, and placing them in a way that allows for easy comparison between cases. It is hard to compare, for example, Figure 4(b), 4(c), 5(b) and 5(c) (as mentioned in line 329) when it requires skipping between pages.

Reediting and typesetting have been done for these figures.

Q13. In particular, a strong disturbance was observed at about 25 seconds from the first subject's time-frequency spectrum diagram. In fact, the PLI is so severe that the spot at that moment overwhelms the image elsewhere in the diagram - > can You show that disturbance in a zoomed in plot before and after filtering? These are always hard to process and the reader might be curious how the methods peformed in this extreme case.

Figure 7 have been added to show the raw and processed signals that mentioned in the question, as well as Figure 8 have been added to show the frequency spectrum of the above signals.

Q14. In fact, could You provide a few plots with interesting cases before and after filtering using Your and other methods? Seeing SNR numbers and spectrum is okay, but this will certainly improve the quality of Your presentation.

Figure 7 and Figure 8 have been added to solve this problem.

Q15. Discussion and conclusions paragraph has different fonts from line 362 to 370.

The modification have been done.

Q16. You should also mention other methods of signal analysis, like EMD and EEMD, which can be applied to non-linear and non-stationary signals. Please consider adding the following sources DOI 10.3390/s22103765, DOI 10.35784/acs-2022-14, DOI 10.3390/s22062176 in the introduction section.

The EMD based methods have been introduced and briefly discussed in section 1.

Reviewer 3 Report

(1)In line of  37 of page 1, "Which makes",  Which used here has some grammar problems.

(2)In line of 56 of page 2, "while the latter two model the PLI as a time-invariant cosine function." has grammar problem. 

(3)In line of 121 of page 4, some math symbols are not defined.

(4)In line of 133 of page 4, there is a grammar problem.

(5)How is to obtain  the  equation (9)?

(6)How is to define the normalized output SNR?The unit of it is dB?

(7)The different performance indexes should be used to evaluate the performance of the proposed scheme.

Many grammar problems are existed in the paper. The writing should be further improved. 

Author Response

Q1. In line of  37 of page 1, "Which makes",  Which used here has some grammar problems.

’Which makes ’ have been modified to ‘ This defect of the SEMG signal makes ’

Q2. In line of 56 of page 2, "while the latter two model the PLI as a time-invariant cosine function." has grammar problem.

This sentence have been modified as ’ while, the PLI signal is modeled as a time-invariant cosine function when using the later two methods.’

Q3. In line of 121 of page 4, some math symbols are not defined.

two modifications have been made:

1.In the second line of the last paragraph of Section 2.1, ‘. Where,  ' have been modified to ‘. Where,  is the time span and ’.

  1. To clear up misunderstandings, the ‘’i’’in Equ (2) have been modified to ‘j’, as the ’i’or ‘’j’’ here is the imaginary unit and ’i’ have been defined in other ways in other equations. Besides, a comment have been added below the Equ (2), as: ’Where, j is the imaginary unit.’.

Q4. In line of 133 of page 4, there is a grammar problem.

I am sorry that i’m not sure where the problem is. As in the line 133 of the Manuscript for Revisions that i downloaded is the section title of 2.3, as ‘2.3. Local CWT/SWT related to the PLI’

Q5. How is to obtain  the  equation (9)?

As the spectrum of raw sEMG signals tends to be distributed continuously, we assume that at any time, within the meaningful frequency range of SEMG, the time-frequency spectrum within any relatively narrow bandwidth obeys the normal distribution, thus, the thrice standard error principle is used to calculate the adaptive threshold in Equ (9).

Q6. How is to define the normalized output SNR?The unit of it is dB?

The normalized output SNR were defined in Equ (23), and it is non-dimensional with a ranges from 0 to 1.

Q7. The different performance indexes should be used to evaluate the performance of the proposed scheme.

In this manuscript, three indexes (the output SNR, the correlation coefficients between and the raw sEMG signals and the processed signals and the root mean square error) have been used to evaluate the performance of the proposed methods in simulated signals, which can be found in Fig 3 and Tab 2-4. The reason for only showing the results of output SNR in the figure is that, the correlation coefficients lacks resolution when it approaches 1, while the range of root mean square is so large that it is difficult to observe the difference when root mean square is small.

Besides, the image difference of the time-frequency spectrum diagram have also been used to evaluate the performance in raw signals, as shown in Fig 4 and Fig 5.

Round 2

Reviewer 1 Report

The manuscript can be accepted 

Reviewer 3 Report

The authors have completed the paper revisions. It can be accepted for publication.

The authors have completed the paper revisions. It can be accepted for publication.